# Identification of Metabolite and Lipid Profiles in a Segregating Peach Population Associated with Mealiness in *Prunus persica* (L.) Batsch

**DOI:** 10.3390/metabo10040154

**Published:** 2020-04-16

**Authors:** Victoria Lillo-Carmona, Alonso Espinoza, Karin Rothkegel, Miguel Rubilar, Ricardo Nilo-Poyanco, Romina Pedreschi, Reinaldo Campos-Vargas, Claudio Meneses

**Affiliations:** 1Centro de Biotecnología Vegetal, Facultad de Ciencias de la Vida, Universidad Andrés Bello, Avenida República 330, Santiago 8370186, Chile; mari.lillo@uandresbello.edu (V.L.-C.); alonso.espinoza.r@gmail.com (A.E.); rothkegel.k@gmail.com (K.R.); miguel.angel.rubilar.romero@gmail.com (M.R.); reinaldocampos@unab.cl (R.C.-V.); 2Escuela de Biotecnología, Facultad de Ciencias, Universidad Mayor, Camino La Pirámide 5750, Huechuraba, Santiago 8580745, Chile; ricardo.nilo@umayor.cl; 3Escuela de Agronomía, Facultad de Ciencias Agronómicas y de los Alimentos, Pontificia Universidad Católica de Valparaíso, Calle San Francisco s/n, La Palma, Quillota 2260000, Chile; romina.pedreschi@pucv.cl; 4FONDAP Center for Genome Regulation, Universidad Andrés Bello, Blanco Encalada 2085, Santiago 87370415, Chile

**Keywords:** chilling injury, mealiness, metabolomics, lipidomics, biomarker

## Abstract

The peach is the third most important temperate fruit crop considering fruit production and harvested area in the world. Exporting peaches represents a challenge due to the long-distance nature of export markets. This requires fruit to be placed in cold storage for a long time, which can induce a physiological disorder known as chilling injury (CI). The main symptom of CI is mealiness, which is perceived as non-juicy fruit by consumers. The purpose of this work was to identify and compare the metabolite and lipid profiles between two siblings from contrasting populations for juice content, at harvest and after 30 days at 0 °C. A total of 119 metabolites and 189 lipids were identified, which showed significant differences in abundance, mainly in amino acids, sugars and lipids. Metabolites displaying significant changes from the E1 to E3 stages corresponded to lipids such as phosphatidylglycerol (PG), monogalactosyldiacylglycerol (MGDG) and lysophosphatidylcholines (LPC), and sugars such as fructose 1 and 1-fructose-6 phosphate. These metabolites might be used as early stage biomarkers associated with mealiness at harvest and after cold storage.

## 1. Introduction

Peaches and nectarines (*Prunus persica* (L.) Batsch) are among the most important temperate fruit crops, with a world production of 24,453,425 t in 2018 [1]. Chile is among the top ten countries for peach and nectarine production, with 319,047 t in 2018 [1]. However, exported fruit quality is compromised due to the long distance to destination markets, since fruits must be maintained at low-temperature storage (0–4 °C) to prolong shelf life by avoiding decay [2]. This prolonged cold storage can negatively affect the sensorial characteristics of peaches and nectarines like flavor and texture, due to the development of physiological disorders known as chilling injury (CI) [2,3]. Chilling injury corresponds to an internal breakdown, limiting the storage life of peaches and nectarines under refrigeration, and includes dry flesh, hard texture, flesh browning, flesh bleeding and mealiness as the main symptoms [2]. Mealiness is a textural disorder that occurs in the mesocarp and can be observed during shelf life, where the affected ripe fruits exhibit a dry grainy feel when chewed [4]. On a cellular level, cell wall pectin metabolism is altered in mealy peach fruit, a gel is formed when pectic substances in intercellular spaces absorb free water, and intercellular adhesion is reduced [5]. At this level, mealiness is associated with a decline in the respiration rate, very low ethylene evolution and reductions in extractable protein of over 50% [5,6,7,8]. Cold storage causes a reduction in ethylene-regulated enzymes, which participate in cell wall enzyme activities, including endo-PG, which are required for normal ripening in melting fresh varieties [5]. At a molecular level, mealiness is associated with the incomplete solubilization of cell wall macromolecules due to an imbalance in the expression of transcripts and enzymatic activity of cell wall-modifying enzymes such as pectin methylesterases (PMEs) and polygalacturonases (PGs) [5,9]. Besides, the dysfunction of cell membranes at low temperatures is considered to be a primary molecular event ultimately leading to the development of CI symptoms [10]. Several studies have linked the activity of polygalacturonase with the incidence of mealiness. PG transcripts cannot accumulate at normal levels as ripe juicy fruits; additionally, an endopolygalacturose that co-localizes with an important QTL affecting characteristics generated by cold damage, such as mealiness and pulp bleeding, has been identified [4,11].

Different genomic studies have been performed to elucidate the mechanisms underlying mealiness. For example, nine candidate genes were identified for peach mealiness from QTL regions, where LG4 contained a cluster for a genetic factor that could regulate mealiness, being a transcription factor and one of the most relevant genes in the regulation of this trait [12]. Additionally, an SNP marker for resistance to chilling injury symptoms has been proposed as a potential candidate gene (QTN) [13]. The presence of genes related to invertase/pectin methylesterase inhibitor (PMEI) was reported in fruit susceptible to mealiness. This gene produces a de-methyl esterification effect, producing a free carboxylic group that alters the charges inside the cells, allowing the addition of polyuronides into a calcium-linked gel structure, which increases the firmness of the cell wall, preventing the incidence of mealiness in the fruits [14].

Metabolomics provides a tool to understand the physiological processes that occur in response to different types of stress, such as chilling injury, where differences in the abundance of these compounds can be used to characterize phenotypes. For instance, in response to cold and other osmotic stresses, plant organisms can change their metabolism and accumulate metabolites like sterols, glucosides, raffinose, arabinoxylans and another soluble sugars [15]. In addition, plants also accumulate other solutes such as glutamic acid, amino acids (alanine, glycine, proline and serine), polyamines (putrescine) and betaines. These molecules, which are often degraded once the stress has passed, are referred to as osmolytes, osmoprotectans or compatible solutes [16,17,18]. Otherwise, lipids are an essential part of biological processes and serve numerous structural and functional roles in these systems inclusive of providing structural molecules for forming cellular membrane bilayers, signaling molecules, energy storage and transport. Advances in lipid detection techniques have been useful for studying the composition of the plasma membrane and how it is remodeled in response to stresses such as cold temperature. For this reason, a metabolome and lipidome approach could be useful for the study of differential metabolic profiles between phenotypes, helping in the search of biomarkers associated with agronomical traits. Thus, this work aims to identify candidate metabolites and lipids involved in the susceptibility to mealiness in *Prunus persica* by GC-MS and UHPLC MS/MS analysis.

## 2. Results and Discussion

### 2.1. Phenotypic Parameters in Response to Chilling Injury

In order to determine the physiological parameters of the V × V nectarine population, we evaluated phenotypic parameters such as firmness [N], background color (I_ad_), soluble solids (°Brix) and juiciness (%). For firmness parameter, background color and soluble solid content (Figure 1A–C), no significant differences were found between contrasting phenotypes for mealiness (juicy and mealy fruits). In order to evaluate the degree of chilling injury in both contrasting individuals from the V × V population, we determined the percentage of juice after cold storage (Figure 1D). In this case, juiciness in mealy fruits was significantly lower (21.56% of juice), compared to juicy fruits (52.68% of juice) as revealed by a t-test (*p* < 0.05), coinciding with the expected characteristics of this physiological disorder.

### 2.2. Comparison of Differentially Accumulated Metabolites and Lipids

A total of 308 metabolites and lipids were initially characterized (Appendix A), and included triglycerides, saturated and unsaturated fatty acids, disaccharides, dicarboxylic acids, amino acids and sugar acids, among others. From these 308 metabolites and lipids, 212 (68.8%) could be mapped to metabolites with unique InChiKeys/Pubchem_ID/SMILES tags, as required by the ChemRICH tool (Figure 2) [19]. Two comparisons were performed: E1 (before cold storage) against E3 (after cold storage) from juicy phenotype fruits (Figure 2A), and E1 against E3 from mealy phenotype fruits (Figure 2B). When assessing the transition from E1 to E3 in fruits that remained juicy, E1 fruits displayed higher levels of triglycerides and lower levels of unsaturated phosphatidylcholines, unsaturated fatty acids and branched chain amino acids compared to E3 (Figure 2A). The same analysis for fruit that would become mealy showed that a decrease in branched chain amino acids was displayed in the E1 to E3 transition. In addition, saturated lysophosphatidylcholines (LPC), diglycerides, sugar acids and hexosephosphates decreased markedly during the E1 to E3 transition in fruit that would become mealy, compared to a null or very slight change in the fruit that remained juicy (Figure 2B).

Lipids are the major components of plasma and endo-membranes, having a structural role to mitigate the impact of temperature variations [20]. Vegetable lipids have a great diversity of structure, ranging from simple lipids as free fatty acids to complex lipids like sphingolipids, and can be classified into approximately eight groups, comprising a wide range of physical and chemical properties that allow their participation in several physiological processes [21]. Changes in lipid profiles in response to low temperature revealed that the composition of membrane lipids changes substantially in plants [22]. A significant increase in unsaturated fatty acids has been reported during cold acclimation [23,24]. In *Arabidopsis thaliana*, the amounts of lysophospholipid species such as lysophospatidylcholine (LPC) and lysophosphatidylethanolamine (LPE) increased in response to cold acclimation at 4 °C or exposure to freezing stress [22,24]. In response to cold, enrichment analysis corresponding to the juicy phenotype showed high levels of triacylglycerol (TAG) (Figure 2A); however, the mealy phenotype showed low amounts of saturated LPC, amino acids, diglycerides, sugar acids and hexosephosphates (Figure 2B). Phospholipids are components of all biological membranes and participate in processes such as signal transduction, cytoskeleton rearrangement and membrane trafficking, and species of lysophospholipids have been described to be involved in different plant processes such as germination, cell expansion and responses to different types of stress [20]. In a previous study related to the mechanisms underlying pit development in blueberries, a disorder associated with cold stress, a lipidomics analysis revealed the involvement of lipid metabolism in such a disorder. For instance, significant increases in DGDG, PA, PS, PG, PI, LPC and LPE were observed at the early stages, followed by a decrease in PC after cold storage, indicating changes in phospholipids associated with cold stress [25]. The exposure of plants to stress conditions like exposure to low temperatures results in the alteration of their metabolism. It has been reported that metabolism is altered by the adjustment or restoration of the catalytic properties of enzymes through regulatory mechanisms in response to cold stress [26,27] and through modifications of metabolic parameters such as the production of some metabolites, osmolytes and phytohormones [28,29,30,31,32,33,34,35,36]. These include sugars, amino acids, organic acids, polyamines and lipids, which eventually assist in cellular protection from cold-induced damage by various mechanisms [28,29,30,31,32,33,34,35,36]. These results suggest that the conditions of susceptibility or tolerance of the fruit are determined at harvest and that fruits of the juicy phenotype probably developed a better adaptation in response to cold than fruits corresponding to the mealy phenotype.

### 2.3. Effect of Cold Storage on the Metabolic and Lipid Profiles

In order to evaluate changes in the metabolic and lipid profiles associated with each phenotype at harvest (E1) and after cold storage (E3), a total of 308 compounds were detected by ALEX-CIS (automated lines exchange-cold injection system), GCTOF MS (Gas Chromatography time-of-flight mass spectrometry) and HPLC-charged surface hybrid column and electrospray (CSH-ESI) quadrupole time of flight with tandem mass spectrometry (QTOF MS/MS). All compounds were found in both phenotypes during E1 and E3 and analyzed by multivariate statistical analysis considering the stage of postharvest and phenotype class (juicy vs. mealy). A Partial Least Squares Regression Discriminant Analysis (PLS-DA) was carried out, using the stage (E1 and E3) for the juicy fruit as the response variable and the identified metabolites as the predictor variables. Considering this, the scores plot for the juicy phenotype explained 46.1% of the variability with two components (Figure 3A). In this projection, the E1 stage (red ellipse) appeared to be separated from the E3 stage (green ellipse). Previous to this PLS-DA analysis, a principal component analysis (PCA) was carried out to confirm discrimination in an unsupervised way of both stages (Appendix A). A Variables Important in Projection (VIP) analysis, with the 25 most important metabolites involved in the discrimination between phenotypes, is displayed in Figure 3B. Additionally, a VIP approach was used to select important features that contribute to this separation (Appendix A). From this, a total of 25 compounds were obtained, and their relative abundances are shown in Figure 3A as a heatmap, where the main compounds that contributed to the separation of the stages of postharvest were trehalose-6-phosphate, pyruvic acid, alpha-ketoglutarate, fumaric acid and nicotinic acid, being present at higher levels at the E1 stage. After cold storage, high relative amounts of amino acids such as proline, valine, alanine, isoleucine, serine, leucine and phenylalanine and other compounds such as 1-kestose and alpha-aminoadipic acid were present in the juicy phenotype. One of the strategies used by many organisms to combat environmental stress is the accumulation of water-soluble compounds, known as compatible solutes or osmolytes. Among the most common metabolites displaying osmoprotection, betaines; sugars such as mannitol, sorbitol and trehalose; and amino acids such as proline have been reported [37]. The accumulation of higher amounts of these compounds in juicy individuals might be indicative of their degrees of resistance to cold. On the other hand, a second PLS-DA analysis was carried out using lipids as predictor variables and the stage as response variable (Figure 3B). Considering this, the score plot of the juicy phenotype explained 66.9% of the variation with the two first components (Figure 3B). Previously, a PCA analysis performed indicated a clear separation of both stages (Appendix A). A total of 25 lipids were identified by VIP analysis (Appendix A) and are shown as a heatmap (Figure 3B). Considering this, the juicy phenotype at the E1 stage appeared to be rich in lipids such as diacylglycerols (DG); phosphatidylcholines (PC) of 32, 33, 34 and 36 carbons; phosphatidylethanolamines (PE) of 34 and 36 carbons; and phosphatidylglycerols (PG) of 32 carbons, and at the E3 stage, high levels of triacylglycerols (TAG), phosphatidylcholines (PC) of 38 carbons and digalactosyldiacylglycerols (DGDG) were observed.

Biological membranes are crucial for the function of cells in living organisms, acting as selective barriers. Their structure and function are not only influenced by membrane proteins, but also by their diverse lipid composition [38]. An increase in the relative content of PC 38:2 was detected only in the juicy phenotype after cold storage (Figure 3B), as seen by Bustamante et al. [39], who also detected high levels of this PC after cold storage. This suggests that high levels of this lipid could be an adaptative response only in the juicy phenotype.

When PLS-DA analysis was performed using the mealy phenotype as a response variable, 54.5% of the variance could be explained with the first two components (Figure 4A). A previous PCA analysis confirmed the discrimination of the samples (Appendix A). VIP analysis (Appendix A) revealed that the E1 stage of postharvest of the mealy phenotype was characterized by higher amounts of trehalose-6-phosphate compared to the juicy phenotype, and of other compounds such as arachidic acid, uridine, nicotinic acid, alpha-ketoglutarate, fumaric acid and pyruvic acid in contrast to the E3 stage, which displayed higher abundances of alanine, phenylalanine, proline, isohexonic acid, hexose-6-phosphate, valine, isoleucine, leucine, alpha aminoadipic acid, O-acetylserine, fructose-1-phosphate, serine, 1-kestose, galactonic acid, fructose-6-phosphate, 3,4-dihydroxycinnamic acid, glycerol-3-galactoside and lyxose (Figure 4A, left panel). For the lipid profile, the PLS analysis (Figure 4B) showed a separation between the stages, as previously confirmed with PCA analysis (Appendix A). The mealy phenotype at the E1 stage appeared to be rich in only two compounds: phosphatidylcholine (PC) of 32 carbons and monogalactosyldiacylglycerol (MGDG) of 36 carbons. At the E3 stage, lipids such as triacylglycerols of 52 and 54 carbons, and diacylglycerols (DG) of 36 carbons were found in higher abundance.

Some metabolites behave in the same way in response to cold. The accumulation of 1-kestose under stress (Figure 3A and Figure 4A) in both phenotypes could be a defense response to low temperatures. This function was demonstrated only in studies of grasses (*Lollium perenne* L.) exposed to drought [40] and cold [41,42]. 1-kestose is a fructan, which may act as a rich polyhydroxy compound capable of preventing water freezing [43] and can contribute to the osmotic potential in a better way than sucrose and oligosaccharides combined [44]. In onions (*Allium cepa*), fructans provide osmotic adjustment during bulb development, and fructan hydrolysis helps to regulate turgor in guard cells [45]. Kestose, belonging to the inulin-type fructans, can protect the structure of the membrane under abiotic stresses [46]. Additionally, high levels of trehalose-6-phosphate were found before cold storage in both phenotypes (Figure 3A and Figure 4A). Trehalose-6-phosphate is an essential signaling metabolite in plants, with an influence on growth and development that competes with other signaling molecules, including phytohormones [47]. It has been proposed to play protective roles under various abiotic stresses, including cold and freezing [48]. The protective effects of trehalose are associated with membrane stability and depression in the phase transition temperature of biomembranes, allowing them to remain amorphous, even under completely dehydrated conditions [49]. These results suggest that trehalose-6-phosphate could be a protector of the cellular membrane, changing its fluidity and thus protecting the fruit from cold stress.

At harvest (E1), there were already differential metabolite profiles between the phenotypes, the principal differences being related to higher contents of beta alanine, pyrophosphate, lactitol and putrescine (Figure 3A) in the juicy phenotype. Putrescine was negatively correlated to susceptibility to CI [50]. The mealy phenotype was instead rich in uridine (Figure 4A). Uridine is a nucleoside, primarily found in sugar beets (*Beta vulgaris*), sugarcane (*Saccharum officinarum*), tomatoes (*Solanum lycopersicum*) and broccoli (*Brassica oleracea*), among others [51]. Previous work reported a higher expression of Uridine Diphosphate-glucose PyroPhosphorylase (UGlcPP) in mealy peach fruits. This enzyme plays a key role in carbohydrate metabolism by catalyzing the reversible conversion of glucose-1-phosphate and UTP to UDP glucose and pyrophosphate, respectively. UDP-glucose is the precursor of several nucleotide sugars and is associated with glycan synthesis [52]. High levels of uridine in the mealy phenotype at E1 suggest that this metabolite might be involved in the alteration of the correct cell wall composition and structure, affecting fruit texture, and could be associated with the mealiness phenotype at early stages.

High levels of amino acids such as alanine, leucine and serine were found only after cold storage in both phenotypes (Figure 3A and Figure 4A). At E3, the mealy phenotype displayed high levels of sugars such as fructose 1 and 6 phosphate sugars (Figure 4A). Besides, in response to cold stress in peach (Figure 3A and Figure 4A), the relative abundance of amino acids such as proline, valine and alanine, among others, increased not only in the mealy phenotype, but also in the juicy phenotype. The high levels of amino acids after cold storage (Figure 3A and Figure 4A) indicate a possible accumulation of metabolites with a protective function in response to chilling stress. It has been reported that proline is a protective molecule which imparts protection against cold stress to many plants, not only maintaining its osmolarity but also acting as a molecular chaperone, thereby stabilizing the structures and functions of important proteins and enzymes [53]. It also protects the plant by maintaining the double stranded structure of genetic material and by up-regulating the oxidative stress machinery [53,54]. A positive correlation between the accumulation of endogenous proline and improved cold tolerance has been found mostly in low temperature-insensitive plants such as barley (*Hordeum vulgare*), rye (*Secale cereal*) and *Arabidopsis thaliana*, among others [30,54,55]. The most probable roles of proline are associated with cytosol acidity, regulation of the NAD+/NADH ratio, the photochemical activity of the photosystem II in thylakoid membranes and a decrease in lipid peroxidation [56,57]. These previous findings suggest that proline acts as protector independently of the phenotype. The relative abundance of alanine increased after cold storage in both phenotypes (Figure 3A and Figure 4A). Previous studies showed that the concentrations of alanine increased in CI-sensitive crops during cold storage [58,59,60], and it was shown that conversion of pyruvate to alanine by glutamate-pyruvate transaminase (GPT) was likely the source of the increased alanine, as levels of pyruvate were found to be very high in chilled cucumbers (*Cucumis sativus*) and eggplants (*Solanum melongena*), while GPT activity was unaffected by the chilling [61].

Others amino acids found in high levels after cold storage in both phenotypes are isoleucine and valine (Figure 3A and Figure 4A). Previous authors [62] also found a remarkable increase in the concentrations of several amino acids, including isoleucine and valine, prior to chilling. These results suggest that the accumulation of these amino acids, including proline, could serve as a priming strategy to protect peach fruits from the CI damage induced by subsequent chilling stress. Alternatively, the accumulating isoleucine and valine may serve as substrates for the synthesis of stress-induced proteins, and these branched-chain amino acids may act as signaling molecules to regulate gene expression [63]. Another hypothesis is that isoleucine and valine could be critical to the maintenance of protein structure and function under cold-stress conditions, because of their unsubstituted aliphatic side chains with branched alkyl groups [64]. Results suggest that the concentrations of amino acids increased in response to cold, independently of the phenotype, which might be indicative that the metabolism of the fruit is focused on protection, increasing the production of metabolites with signaling functions and that participate in the synthesis of proteins related to secondary metabolism. 

### 2.4. Metabolites and Lipids Associated with Cold Stress and Possible Candidate Biomarkers of Mealiness

Furthermore, the relative abundances of the top metabolites with differential abundance between the phenotypes in another two individuals from each phenotype (Figure 5 and Figure 6) were assessed and correlated with those in the previous individuals (Appendix A). High levels of 2-hydroxyglutaric acid, tryptophan and threonine (Figure 5A) after the cold storage were found, as well as high levels of some lipids (Figure 5B) such as PG (34:2) and DGDG (36:4) at the E3 stage in juicy phenotype. DGDG is a bilayer lipid that may increase membrane stability [39,65], suggesting a possible cold tolerance in these individuals. Additionally, PE (34:2) was present in higher relative amounts at E1 only in the juicy phenotype.

The relative abundance of the top metabolites in the mealy phenotype (Figure 6A) revealed high levels of arachidic acid at the E1 stage and higher levels of fructose 1 and 6 phosphate at the E3 stage. It has been extensively reported that high contents of sugars alleviate CI symptoms in peach fruit, because carbohydrates may serve as osmoregulators and cryoprotectans, contributing to membrane stability [66,67,68]. Besides, carbohydrates may act as scavengers of reactive oxygen species, and sugar metabolism might provide reducing power to the ascorbate-glutathione cycle protecting cells against chilling stress [49,67,68]. Arachidic acid participates in the biosynthesis of unsaturated fatty acids. Previous authors proposed fructose 6 phosphate and arachidic acid as physiological markers for incipient chilling injury in tomato [68]. If the levels of arachidic acid are low, the biosynthesis of unsaturated lipids becomes deficient, thus increasing the rigidity of the plasma membrane of mealy individuals. At the lipidome level, high amounts of MGDG in the mealy phenotype at the E1 stage were observed, decreasing at E3 stage (Figure 6B). MGDG is a non-bilayer lipid that can severely destabilize membranes. It has been proposed that an increase in the ratio of bilayer to non-bilayer-forming membrane lipids results in the stabilization of membranes during freezing [39,67]. The differential levels of MDGD in the mealy phenotype suggest that the membrane stability is altered in response to cold. This phenotype is inverse in the juicy phenotype, where high levels of DGDG at E3 stage were found, suggesting an increase in the membrane stability (Figure 5B). In addition, it has been shown that an increase in the content of di-unsaturated species of PC in the rye protoplast leads to an increased tolerance of the plasma membrane against freezing [39,69]. Recent studies under cold stress have reported that besides the increase in the degree of fatty acid unsaturation, the main constituent component of chloroplast-specific lipids is also susceptible; a decrease in monogalactosyldiacylglycerol (MGDG) has been shown [70,71,72]. Besides, a notable increase in phosphatidic acid (PA), lysophosphatidylethanolamine (LPE) and lysophosphatidylcholine (LPC) has been reported in response to low temperature [72], with LPC being more abundant in the mealy phenotype after cold exposure. All of these observed changes in response to cold stress suggest that the plasma membrane undergoes remodeling, probably trying to decrease unsaturated lipids like MGDG, whilst increasing metabolites that participate in the biosynthesis of this type of lipid (e.g., arachidic acid), and increasing other stabilizing lipids like DGDG to make the plasma membrane more rigid. These changes are only visible in the mealy phenotype, which could be indicative that these individuals must adapt their metabolism in response to chilling exposure and thus need to regulate membrane permeability. Having observed these differential responses in metabolism, the metabolites that changed in abundance should be further studied to confirm their roles as potential biomarkers for CI.

## 3. Materials and Methods

### 3.1. Plant Material and Phenotyping

Two siblings from the nectarine segregating population (V × V) were selected, considering contrasting levels of mealiness susceptibility or juice content after cold storage. The V × V population was obtained from the self-pollination of the “Venus” nectarine variety, which is located on INIA Rayentué in Chile (34°32′14′′ S, 70°83′44′′ W). This population is freestone and melting and has yellow flesh. Fruits were harvested between mid-December and late January, when their firmness was around 53 N. Two V × V siblings, contrasting in mealiness were selected (one sibling juicy and another mealy). A group of harvested fruits (E1) was ripened at 20 °C for eight days until their firmness was around 4–8 N, corresponding to the E2 stage (commercial firmness). Another group of fruits was stored at 0 °C for 30 days, corresponding to the E3 stage of postharvest. After these 30 days, fruits were kept at 20 °C for seven days until the firmness was around 7 N, corresponding to the E4 stage of postharvest. As biological replicates, five fruits from selected contrasting siblings were evaluated at the different postharvest stages. In E1, the parameters evaluated were weight, firmness, background color and soluble solids (SS). For E2 and E4, the parameters evaluated were firmness and juiciness (% of juice), while in E3, only firmness was measured. Flesh firmness was assessed on two paired sides of each fruit after exocarp removal, using a Fruit Pressure Tester fitted with an 8 mm diameter plunger (FT327 and FT011, EFFEGI). The soluble solids content (SS) was determined using a digital refractometer (WINEline HI96811, HANNA, RI, USA). The background color was evaluated using a DA meter (I_ad_) (Sintéleia FRM01-F, Bologna, Italy), and juiciness was evaluated as previously described by Infante et al. [73]. Briefly, with the exocarp previously removed, a piece of mesocarp was taken from each side of the fruit and placed on a previously massed absorbent paper. Then, this paper was folded over the piece of mesocarp and weighed with a balance. The sample was covered with two previously massed absorbent papers, and then a mechanical stress was generated on the mesocarp. Finally, these papers were weighed, and the percentage of juice was obtained by the weight difference of the papers. Finally, fruits with 30% or less of juice content were considered as mealy [74].

### 3.2. Metabolite and Lipid Extraction

Samples from stages E1 (before cold storage) and E3 (after cold storage) were used to evaluate the metabolite and lipid profiles of both contrasting phenotypes. Total metabolites were extracted as described by Fiehn et al. [75], where 20 mg of frozen tissue corresponding to mesocarp were placed in a 1.5 mL microtube containing 1 mL of pre-cooled extraction buffer (5:2:2, (v/v/v) of degassed methanol, chloroform and water) and macerated at 1750 rpm for 1 min at 4 °C using a ball mill grinder (MM301, Retsch Corp., Hann, Germany). Tubes were centrifuged for 2 min at 14,000× *g* and 4 °C using a bench-top centrifuge (Eppendorf 5415D, Hamburg, Germany), discarding the precipitate (plant debris). Five hundred milliliters of supernatant were transferred into a new 1.5 mL microtube and dried using a speed vacuum concentrator (Centrivap cold trap concentrator, Labconco, Kansas, MO, USA). For metabolite derivatization, 20 μL of methoxyamine solution (Sigma-Aldrich, St. Louis, MO, USA) containing 20 mg/mL of pyridine (Sigma-Aldrich, St. Louis, MO, USA ) were added to the dried sample tubes and shaken for 90 min at 30 °C. To each tube, 91 μL of MSTFA (*N*-Methyl-*N*-(trimethyl-d_9_-silyl)trifluoroacetamide) (Sigma-Aldrich, St. Louis, MO, USA) and 10 μL of FAME (Fatty Acid Methyl Ester) marker (Supelco C8-C24, Sigma-Aldrich, St. Louis, MO, USA) were added and shaken for 20 min at 37 °C. Each prepared sample was transferred to an autosampler vial with a micro-insert.

Lipid extraction was carried out as described by Matyash et al. [76]. Briefly, 20 mg of the organic phase was extracted with 1.5 mL of methanol and 5 mL of methyl-tertiary butyl ether (MTBE). Tubes were vortexed for 20 s and centrifugated for 2 min at 14,000× *g*. Then, the organic phase was separated and was reconstituted in 65 μL of a solution 9:1 (v/v) of methanol and toluene, with N-cyclohexyl-N′-dodecanoic acid urea (CUDA) as an internal standard. Each prepared sample was transferred to a vial with a micro-insert.

### 3.3. Metabolite and Lipid Analysis

Metabolite analysis was performed by gas chromatography-mass spectrometry (GC-MS) and carried out by the NIH West Coast Metabolomics Center. Data were obtained using the protocol described by Fiehn et al. [75]. Briefly, 0.5 μL of sample were injected in splitless mode into a GC-MS system-mass detector (Leco Pegasus IV mass spectrometer, LECO Corp., St. Joseph, MI, USA). The mass spectrometry parameters used corresponded to mass resolution at 17 spectra per second from 80–500 Da at a −70 eV ionization energy and 1800 V detector voltage with a 230 °C transfer line and a 250 °C ion source. Initially, the injection temperature was 50 °C and ramped to 250 °C by 12 °C per second, while the column temperature started at 50 °C for 1 min and ramped to 330 °C. The GC column was a Rtx-5Sil MS (Restex) of 30 m length and 0.25 mm of internal diameter, with 0.25 μm film made of 95% dimethyl/5% diphenylpolysiloxane with a constant flow of 1 mL/min helium gas as the mobile phase. The oven temperature program was set to 50 °C for 1 min and then ramped to 20 °C for 1 min and maintained at 330 °C for 5 min. Automatic mass spectral deconvolution was performed with peak detection of the GC spectrum using the BinBase algorithm (rtx5). The BinBase algorithm was set as the validity of chromatogram (>10 peaks with intensity >10^7^ counts s^−1^), unbiased retention index marker detection (MS similarity >800, validity of intensity range for high m/z marker ions), and the retention index calculation by fifth order polynomial regression. Spectra were cut to 5% base peak abundance and matched to database entries from the highest to lowest abundant spectra using the following matching filters: retention index window ± 2000 units (equivalent to about ± 2 s retention time), validation of unique ions and apex massed (unique ion must be included in apexing masses and present at >3% of base peak abundance), and mass spectrum similarity must fit criteria dependent on peak purity and signal/noise ratios and a final isomer filter. All thresholds reflect settings for ChromaTop v2.32. Quantification was reported as peak height using the unique ion as default, unless a different quantification ion was manually set in the BinBase administration software BinView.

Lipid analysis was carried out by the NIH West Coast Metabolomics Center and performed by Ultra High-Pressure Liquid Chromatography (UHPLC) with charged surface hybrid column and electrospray (CSH-ESI). The detector employed was a quadrupole time of flight with tandem mass spectrometry (QTOF MS/MS). Data were acquired by injecting 3 μL of sample into a column Waters Acquity UPLC CSH C18 (100 mm length × 2.1 mm internal diameter and 1.7 μm particles) with a mobile phase A of 60:40 (v/v) acetonitrile/water, 10 mM ammonium formiate and 0.1% of formic acid; and a mobile phase B of 90:10 (v/v) isopropanol:acetonitrile, 10 mM ammonium formiate and 0.1% of formic acid. The column temperature was 65 °C, the flow rate was 0.6 mL/min and the injection temperature was 4 °C. Data were processed in an untargeted manner by Agilent′s software MassHunter Qualitative in order to find peaks. Peak features were imported into MassProfilerProfessional for peak alignment and filtering and the detection of target lipids. Using the MS/MS information and the lipidBlast library, it was possible to identify lipids with manual confirmation. Then, the data were quantified by MassHunter Quantitative software using the unique retention time and the mass/z identification.

### 3.4. Metabolic Pathway Assessment

In order to determine enrichment in metabolites when comparing conditions, the first step to characterize pathways, the ChemRICH tool [19] was used following the default conditions. Student′s t-tests were performed to evaluate differences between the pairs of conditions to be assessed. Metabolites that had no or duplicated Pubchem ID, or that had no InChiKeys, were excluded from this analysis.

### 3.5. Statistical Analysis

Partial Least Squares Regression-Discriminant Analysis (PLS-DA) and Principal Component Analysis (PCA) were performed on the normalized data, using the dataset with results from GC-MS and UHPLC MS/MS analysis, using MetaboAnalyst 4.0 (Xia Lab, McGill University, Quebec, QC, Canada). PLS-DA analysis was used with metabolites as predictor variables, while the phenotype and stage of postharvest were response variables. To assign an equal variance, all variables were mean centered and weighted by standard deviation. To select important features, Variables Important in Projection (VIP) scores were employed to filter PLS results. These data were analyzed using Student′s t-test statistical tools (*p* < 0.05) to identify compounds with significant differences between E1 vs. E3, and mealy vs. juicy.

Additionally, in order to validate the metabolites and lipids found in each phenotype, two more individuals with five biological replicates from each phenotype were used for evaluating the metabolites and lipids that showed a differential abundance in the multivariate analysis. These data were analyzed using Student′s t-test statistical tools (*p* < 0.05) in order to identify statistical differences between the stages of postharvest. In order to correlate differences between samples, we performed a hierarchical clustering analysis using Spearman’s rank correlation and single linkage as the clustering algorithm.

## 4. Conclusions

Our results indicate that fruits from both phenotypes (mealy and juicy) present differences in metabolism before cold storage, probably further determining cold resistance. After cold storage (E3), metabolites related to primary metabolism such as sugar metabolism were partially impaired. Most of the metabolites present in lower amounts in the mealy phenotype were related to membrane stability (MGDG and PG). However, those metabolites present in higher amounts were related to cold stress (sugars and LPC). These metabolites and lipids could be used as early potential biomarkers for CI in support of farmers, detecting the possible appearance of mealiness before sale and exportation.

## Figures and Tables

**Figure 1 metabolites-10-00154-f001:**
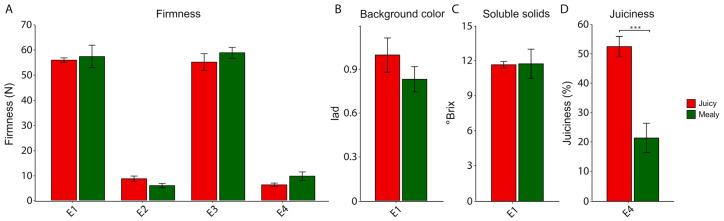
The physiological parameters of nectarine fruits from the V × V population: (**A**) firmness, (**B**) background color, (**C**) soluble solids and (**D**) juiciness. These parameters were measured in mealy and juicy fruits during 2016 and 2017 for contrasting individuals of the V × V population for mealiness (three siblings resistant to mealiness and three siblings susceptible to mealiness). Values are the means of five biological replicates (five fruits) and the bars represent the standard deviation. (*) indicates significant differences between both phenotypes, according to a t-test at the 0.05 significant level.

**Figure 2 metabolites-10-00154-f002:**
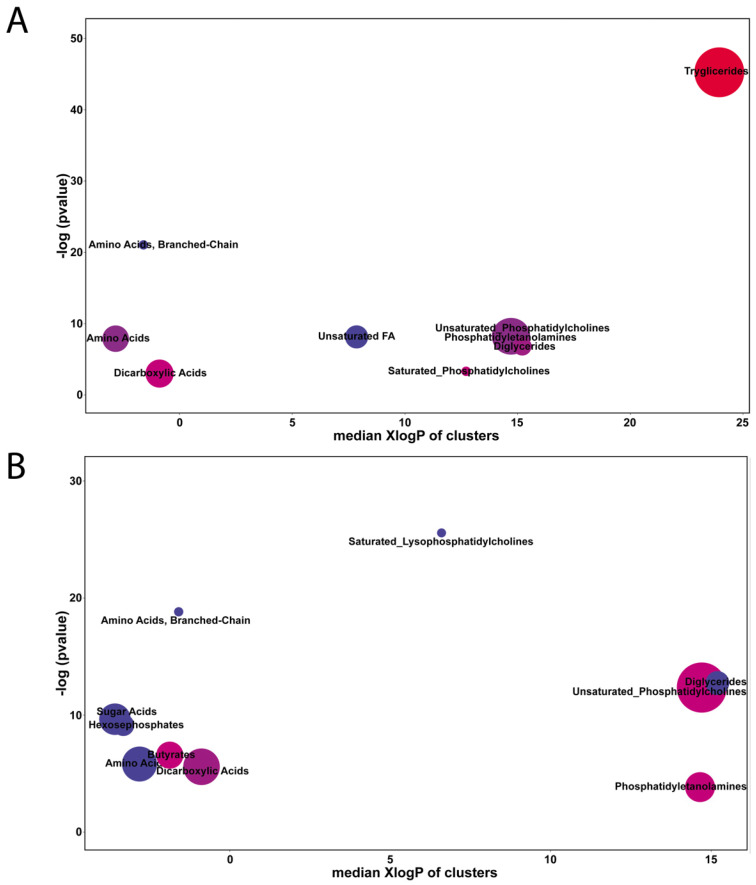
A comparison of E1 vs. E3 in terms of differentially accumulated metabolites in juicy (**A**) and mealy (**B**) fruit. Chemical Similarity Enrichment (ChemRICH) analysis was used to identify metabolites with significant alterations in their abundances during the transition from firm (E1) to cold stored (E3) fruit. Each circle reflects a significantly altered cluster of metabolites. The circle sizes represent the total number of metabolites in each cluster set. The circle color scale shows the proportion of increased (red) or decreased (blue) compounds. Purple-color circles have both increased and decreased metabolites. The *X*-axis correlates clusters in terms of their average lipophilicity (XlogP- octanol/water partition). On the *Y*-axis, the enrichment *p*-values are given by the Kolmogorov-Smirnov-test.

**Figure 3 metabolites-10-00154-f003:**
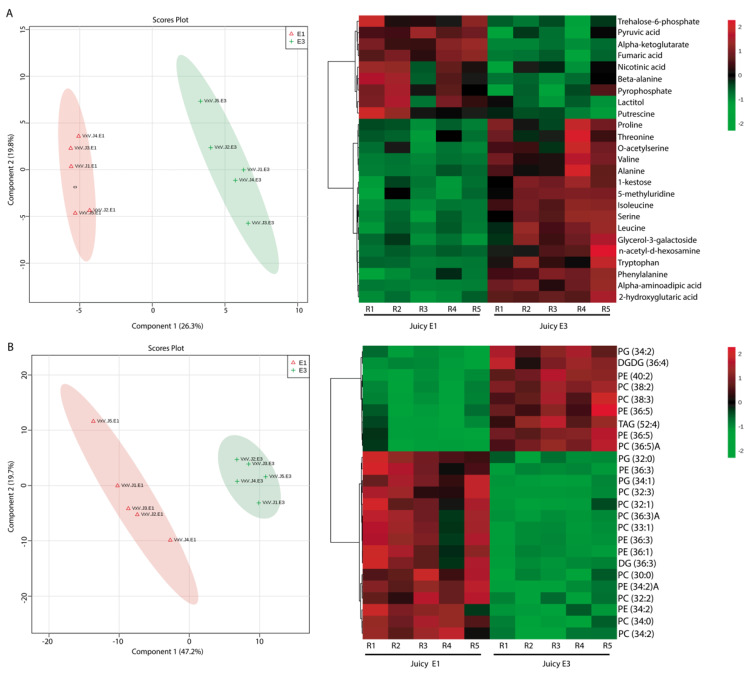
The metabolite and lipid profiles of nectarines at the E1 and E3 stages of the juicy phenotype. (**A**) Multivariate analysis for metabolite profiles and (**B**) Multivariate analysis for lipid profiles. The left panel shows the Partial Least Squares Regression Discriminant Analysis (PLS-DA). The first component (Component 1) is shown on the x axis and the second component (Component 2) is shown on the y axis. The detected metabolites and lipids were employed as predictor variables, and the stage of postharvest as a response variable. In the right panel, a heatmap analysis for metabolite and lipid profiles are displayed, based on the top 25 compounds identified by PLS-DA Variables Important in Projection (VIP). The columns represent biological replicates for each stage (E1: harvest, E3: after cold storage). The similarity measure employed to group the different features was calculated based on Euclidean distance and Ward′s linkage.

**Figure 4 metabolites-10-00154-f004:**
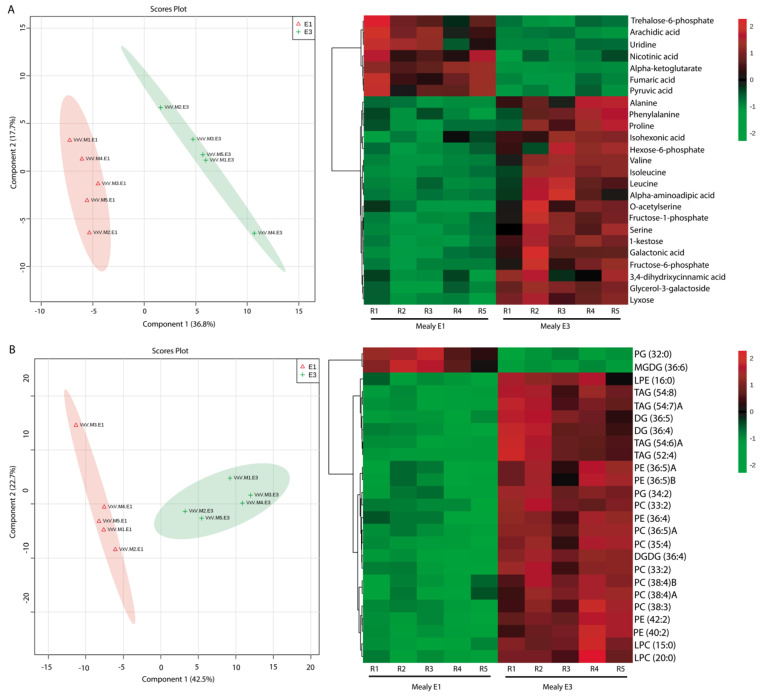
The metabolite and lipid profiles of nectarine at the E1 and E3 stages of the mealy phenotype. (**A**) Multivariate analysis for the metabolite profile and (**B**) Multivariate analysis for the lipid profile. The left panel shows the Partial Least Squared Regression Discriminant Analysis (PLS-DA). The first component (Component 1) is shown on the x axis, and the second component (Component 2) is shown on the y axis. The detected metabolites and lipids were employed as predictor variables, and the stage of postharvest as a response variable. In the right panel, a heatmap analysis for the metabolite and lipid profiles is displayed based on top 25 compounds identified by PLS-DA VIP. The columns represent biological replicates for each stage (E1: harvest, E3: after cold storage). The similarity measure employed to group the different features was based on Euclidean distance and Ward´s linkage.

**Figure 5 metabolites-10-00154-f005:**
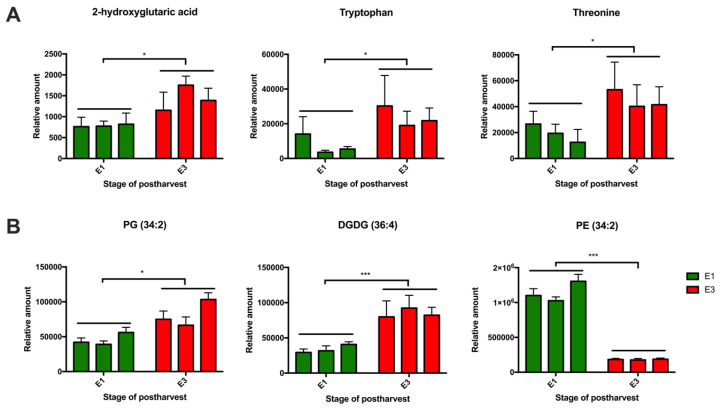
The relative amounts of metabolites (**A**) and lipids (**B**) at the E1 and E3 stages, using three V × V siblings resistant to mealiness (>30% of juice content) from a peach segregating population. Each column represents the average of five biological replicates (n = 5), and the bars are the standard deviations. Green bars show the E1 stage (harvest) and red boxes show the E3 stage (harvest + 21 days at 0 °C). Statistical analysis was performed by t-tests using *p* < 0.05.

**Figure 6 metabolites-10-00154-f006:**
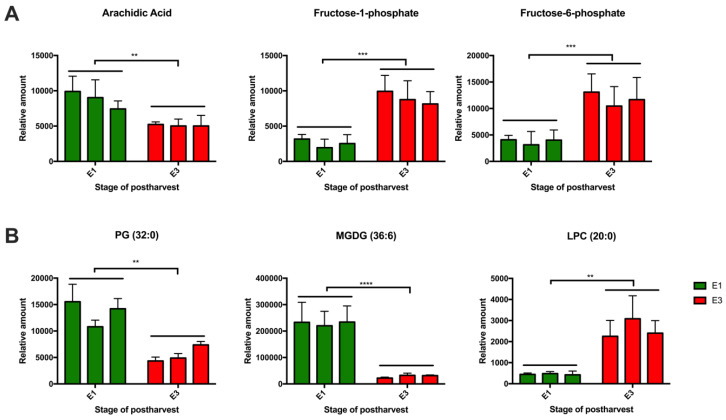
The relative amounts of metabolites (**A**) and lipids (**B**) at the E1 and E3 stages, using three V × V siblings susceptible to mealiness (<30% of juice content) from a peach segregating population. Each column represents the average of five biological replicates, and the bars are the standard deviations. Green boxes show the E1 stage (harvest), and red boxes show the E3 stage (harvest + 21 days at 0 °C). Statistical analysis was performed by using t-tests with *p* < 0.05.

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
