# Peer review of "Identification of Metabolite and Lipid Profiles in a Segregating Peach Population Associated with Mealiness in Prunus persica (L.) Batsch"

_metabolites, 2020, doi:10.3390/metabo10040154_

Round 1

Reviewer 1 Report

The manuscript describes metabolome profiles to investigate the difference between two siblings at harvest and after cold storage. From the view point of metabolomics application, this is an interesting. The results sound scientifically correct. However, the manuscript lacks some important information. There is no explanation of the 308 compounds identification. Did you confirm that they are right based on what kind of evidence without any standard compound? MS fragmentation, retention time, and MS/MS data are very important in metabolomic analysis.

 Abstract: this section should be reformulated to provide main findings, novelty and implications. Provide your opinions to support the significance of the work

 Line 64: Metabolome refers to the complete set of small-molecule, including lipid composition.

Are metabolomics and lipidomics different?

 Figure 1-6: The figures have low resolution.

 Line 107: Explain E1 and E3.

 Line 136: What is ALEX-CIS?

 Line 156: Did you detect trehalose-6-phosphate using GC-TOFMS? Did you confirm by co-injection of commercial standard?

 I suggest to refer to COSMOS standards of reporting in metabolomics.

Author Response

Response to Reviewers

The authors thank the reviewers for their time and dedication in reviewing the manuscript. We agree with the valuable comments and proceed to address your reviews. In addition, all changes have been incorporated in the final version of this manuscript and can be tracked down highlighted in yellow.

Reviewer 1

The manuscript describes metabolome profiles to investigate the difference between two siblings at harvest and after cold storage. From the viewpoint of metabolomics application, this is an interesting. The results sound scientifically correct. However, the manuscript lacks some important information. There is no explanation of the 308 compounds identification. Did you confirm that they are right based on what kind of evidence without any standard compound? MS fragmentation, retention time, and MS/MS data are very important in metabolomic analysis.

ACTION: We performed a metabolite and lipid analysis through NIH West Coast Metabolomics Center service. They provide the retention index of each compound, in addition to its mass/z and fragmentation pattern of each compound. Both parameters were used to identify the compounds used in this manuscript. In addition, standards for the identification of some compounds are used by the NIH West Coast Metabolomics Center. Identification parameters are detailed in the materials and methods section. Internal standards corresponding to different fatty methyl esters are also used. All this information has been included in the final version of this manuscript in supplementary tables 1 and 2.

Reviewer 1

 Abstract: This section should be reformulated to provide main findings, novelty, and implications. Provide your opinions to support the significance of the work.

ACTION: As suggested by the reviewer, we have edited the most relevant findings and added in the abstract, the most significant findings and future implications.

Reviewer 1

Line 64: Metabolome refers to the complete set of small-molecule, including lipid composition.

Are metabolomics and lipidomics different?

ACTION: We appreciate the reviewer for her/his question. Although there are no differences between metabolomics and lipidomics from the point of view that you are addressing, we use these two different terms to distinguish between the two based on the experimental strategy used for the detection. Following the point raised by the reviewer, the manuscript has been modified on line 66.

Reviewer 1

 Figure 1-6: The figures have low resolution.

ACTION: Quality of figures have been improved. All figures have been modified to 1200 pixels per inch to have the right resolution.

Reviewer 1

Line 107: Explain E1 and E3.

ACTION: The requested explanation of E1 and E3 has been included in the final version of this manuscript as requested by the reviewer on line 109.

Reviewer 1

Line 136: What is ALEX-CIS

ACTION: The acronym ALEX-CIS has been fully spelled out in the final version of this manuscript on line 164.

Reviewer 1

 Line 156: Did you detect trehalose-6-phosphate using GC-TOFMS? Did you confirm by co-injection of a commercial standard? I suggest to refer to COSMOS standards of reporting in metabolomics.

ACTION: We appreciate the comment. This compound was not detected by a commercial standard; however, its identification is based on its retention index, mass/z of quantification ion and fragmentation pattern. Supplementary tables are provided will all this relevant information for the compounds of interest.

Reviewer 2 Report

In the manuscript the author compared the metabolomics and lipidomics of juicy and mealy siblings of a peach population before and after cold treatment. My concerns are as follows.

Major:

1 With no doubt, the juiciness parameter is the most important phenotype against CI in the manuscript. It will be great if this parameter is measured in E1-4 stages for both siblings in Fig.1.

2 There is no explanation of what "median XlogP of cluster" stands for in the Fig.2 legend. I think maybe PCA is more appropriate to show the overview of metabolomics and lipidomics data.

3 The discussion about the significantly changed metabolite is thin and scatted. I didn't see a big picture of the stress response of the two peach siblings in a metabolism level.

4 A complete list of identified metabolites and lipids should be supplied.

5 The conclusion in Line 447-450 can be drawn only for the mealy sibling according to Fig.6. Besides, I'm not sure why the authors were looking biomarkers for peach mealiness. Isn't that a simple bite and taste efficient and costless enough for the purpose?

Minor:

1 The texts in the figures are too small especially in Fig.2 with colored background.

2 The term "LPC" and "LPE" in Line 129 and "TAG" in Line 131 were first referred without full names.

3 Switch the "left" and "right" panel in the legend of Fig.3 and 4.

Author Response

Response to Reviewers

The authors thank the reviewers for their time and dedication in reviewing the manuscript. We agree with the valuable comments and proceed to address your reviews. In addition, all changes have been incorporated in the final version of this manuscript and can be tracked down highlighted in yellow.

Reviewer 2

In the manuscript, the author compared the metabolomics and lipidomics of juicy and mealy siblings of a peach population before and after cold treatment. My concerns are as follows.

Major:

 Reviewer 2

With no doubt, the juiciness parameter is the most important phenotype against CI in the manuscript. It will be great if this parameter is measured in E1-4 stages for both siblings in Fig.1.

ACTION: We value this comment. The evaluation of juice content can only be measured at ripe fruit stages which correspond to E2 and E4 stages. Although we can evaluate the juice content in the E2 stage, the main differences related to the chilling injury can be seen only after cold storage, which corresponds to E4 stage.

Reviewer 2

1.There is no explanation of what "median XlogP of cluster" stands for in the Fig.2 legend.

ACTION: We apologize for not having explicitly informed the meaning of "median XlogP of the cluster". We have added this information in Figure 2 legend in the final version of this manuscript as shown below (highlighted).

Figure 2. Comparison of E1 vs E3 differentially accumulated metabolites in juicy (A) and mealy (B) fruit. Chemical Similarity Enrichment (ChemRICH) analysis was used to identify metabolites with a significant alteration in their abundance during the transition from the firm (E1) to cold stored (E3) fruit. Each circle reflects a significantly altered cluster of metabolites. Circle sizes represent the total number of metabolites in each cluster set. The circle color scale shows the proportion of increased (red) or decreased (blue) compounds. Purple-color circles have both increased and decreased metabolites. X-axis correlates clusters in terms of their average lipophilicity (XlogP - octanol/water partition). Y-axis, enrichment p-values, are given by the Kolmogorov-Smirnov-test.

1. I think maybe PCA is more appropriate to show the overview of metabolomics and lipidomics data.

ACTION: We appreciate the comments from the reviewer since it highlights the necessity to explain why we performed both chemical similarity enrichment analysis and principal component analysis to assess our metabolome data. In our case, we show results of PLS-DA analysis in the manuscript and as supplementary figure, we show results of principal component analysis (score and biplots) for the different scenarios.

The biological interpretation of the observed changes in a metabolome dataset can be derived from different kinds of analysis. The results shown in Figure 2 derive from the enrichment analysis of unique and non-overlapping sets of related molecules, performed by the program ChemRICH (Barupal and Fiehn, 2017). On the other hand, multivariate statistics, like principal component analysis, exploits correlations between any metabolites to obtain global metabolic phenotypes and to discriminate between samples in an unsupervised way (Barupal et al., 2018). Thus, both analyses are complementary, since the first one focuses on the kinds of metabolites displaying high relevance on the differences shown by the samples assessed, whereas the second allows discriminating groups of samples using the information of all metabolites at the same time.

References:

Barupal DK, Fiehn O. Chemical Similarity Enrichment Analysis (ChemRICH) as an alternative to biochemical pathway mapping for metabolomic datasets. Sci Rep. 2017 Nov 6;7(1):14567.

Barupal DK, Fan S, Fiehn O. Integrating bioinformatics approaches for a comprehensive interpretation of metabolomics datasets. Curr Opin Biotechnol. 2018 Dec; 54:1-9.

Reviewer 2

The discussion about the significantly changed metabolite is thin and scatted. I didn't see a big picture of the stress response of the two peach siblings in a metabolism level.

ACTION: We appreciate this comment, the discussion about significantly changed metabolites has been improved and the changes were highlighted in yellow in the results and discussion section.

Reviewer 2

A complete list of identified metabolites and lipids should be supplied.

ACTION: A complete list of the identified metabolites was added to the results as a supplementary table, containing relevant information such as retention index, quantitative ion and mass spectrum for metabolomics data and retention time and m/z data for lipidomics.

Reviewer 2

The conclusion in Line 447-450 can be drawn only for the mealy sibling according to Fig.6. Besides, I'm not sure why the authors were looking biomarkers for peach mealiness. Isn't that a simple bite and taste efficient and costless enough for the purpose?

ACTION: The purpose of our work is to identify metabolites that could serve as early biomarkers, thus before sale and exportation. The mealy phenotype could be detected at early stages before cold storage, thus reducing the economic losses related to chilling injury. We made changes to the conclusion and highlighted in yellow on line 493.

Reviewer 2

Minor:

The texts in the figures are too small especially in Fig.2 with a colored background.

ACTION: Figure 2 font size has been increased in order to make the figure texts more visible.

The term "LPC" and "LPE" in Line 129 and "TAG" in Line 131 were first referred without full names.

ACTION: Full names of all acronyms have been spelled out when first used. These changes can be tracked down in the final version of the manuscript on line 138.

Switch the "left" and "right" panel in the legend of Fig.3 and 4.

ACTION: We regret the confusion; we have changed the names in the legend.

Round 2

Reviewer 2 Report

I think my concerns are properly addressed in this version by the author.